# A TGFB2/TNF-induced *in vitro* model of proliferative vitreoretinopathy (PVR) using ARPE-19 cells confirms nicotinamide as an inhibitor of EMT and VEGFA secretion

Yuqing Huang, Roland Meister, Migle Lindziute⬥, Maximilian Binter⬥, Jan Tode, Carsten Framme, Heiko Fuchs⬥*

Department of Ophthalmology, University Eye Hospital, Hannover Medical School, Hannover, Germany

* fuchs.heiko@mh-hannover.de

## Abstract

Proliferative vitreoretinopathy (PVR) is a vision-threatening fibrotic retinal disorder characterized by the epithelial-mesenchymal transition (EMT) of retinal pigment epithelial (RPE) cells. In this study, we established a pathophysiologically relevant *in vitro* model by co-stimulating ARPE-19 cells with transforming growth factor beta 2 (TGFB2) and tumor necrosis factor-alpha (TNF), referred to as 'TNT', and evaluated the anti-fibrotic and anti-angiogenic effects of Nicotinamide (NAM), a vitamin B3 derivative previously reported to counteract fibrosis in various disease models. Confluent ARPE-19 cells were treated with TGFB2, TNF, or TNT for up to six days. EMT progression was assessed via immunocytochemistry, Western blotting, and collagen gel contraction assays. Live-cell imaging (LCI) combined with Hoechst 33342 nuclear staining and automated tracking using Fiji/TrackMate enabled real-time analysis of cell migration and multicellular aggregation. VEGFA secretion was quantified by ELISA. TNT stimulation induced synergistic EMT-like features, including cell elongation, directional migration, extracellular matrix (ECM) remodeling, gel contraction, and formation of multicellular aggregates. TrackMate-based analysis revealed coordinated nuclear migration under TNT conditions. VEGFA secretion was significantly elevated at early time points. NAM co-treatment reduced ECM protein expression (FN1, COL1A1), attenuated migration and contraction, and significantly lowered VEGFA release. This TNT-based ARPE-19 model represents a robust, live-cell-compatible *in vitro* system that mimics both fibrotic and pro-angiogenic aspects of PVR. It allows real-time assessment of EMT progression and is suitable for screening anti-fibrotic compounds. Our findings suggest that Nicotinamide mitigates both fibrotic and angiogenic responses in this model and may hold therapeutic potential for fibrotic retinal diseases.

**Data availability statement:** All relevant data are within the manuscript and its Supporting information files. Raw data for Figures are listed in Supplementary Table ST1 raw data.

**Funding:** The author(s) received no specific funding for this work.

**Competing interests:** The authors have declared that no competing interests exist.

## Introduction

Proliferative vitreoretinopathy (PVR) is a severe, pathological fibrotic response that can develop as an aberrant consequence of retinal detachment repair surgery or ocular trauma. Although it occurs in approximately 5–10% of primary rhegmatogenous retinal detachment (RRD) cases, PVR remains the leading cause of surgical failure in retinal repair [1]. Clinically, it is characterized by the formation of fibrotic membranes on the epiretinal and subretinal surfaces, as well as within the vitreous cavity, leading to retinal traction, contraction, and ultimately recurrent detachment [2]. Despite extensive research, the exact mechanisms underlying PVR pathogenesis remain incompletely understood.

Currently, it is widely accepted that PVR involves complex processes including epithelial-mesenchymal transition (EMT) of retinal pigment epithelial (RPE) cells, inflammatory responses, extracellular matrix (ECM) remodeling, and subsequent fibrotic membrane formation [3]. During EMT, RPE cells lose epithelial characteristics such as polarity and intercellular adhesion, undergo cytoskeletal reorganization, and gain migratory and contractile properties, accompanied by the secretion of ECM components like fibronectin and type I collagen. Collectively, these alterations contribute significantly to membrane formation and tractional retinal detachment [3,4].

Additionally, EMT in PVR is associated with elevated secretion of pro-angiogenic factors, notably vascular endothelial growth factor A (VEGFA), which plays a key role in pathological neovascularization. Indeed, increased VEGFA levels have been documented across various vitreoretinal diseases, and anti-VEGF therapies have become a standard treatment approach in clinical ophthalmology [5,6]. Nevertheless, the precise role of EMT-driven VEGFA secretion specifically within PVR pathogenesis remains poorly characterized and warrants further investigation.

*In vitro* models of PVR-associated EMT typically utilize immortalized or primary RPE cells, and EMT is commonly induced by exogenous cytokines such as transforming growth factor-β (TGFB2), tumor necrosis factor-α (TNF), interleukin-6 (IL6), fibroblast growth factor-2 (FGF2), Gremlin, and Factor Xa [7,8]. Among these, TGFB2, the predominant isoform in ocular tissues, is a well-established EMT inducer that activates both Smad-dependent and non-Smad pathways [9]. TNF, a key inflammatory mediator secreted by retinal microglial cells, can synergistically enhance TGFB2-induced signaling, thereby accelerating EMT, fibrosis, and pathological tissue remodeling [10–12]. While many EMT studies rely on single-cytokine stimulation, cytokine profiling of vitreous humor from PVR patients reveals a complex milieu of inflammatory and fibrotic mediators [13]. This suggests that single-factor models may not sufficiently replicate the multifactorial nature of EMT *in vivo*. To address this limitation, we established a more pathogenetic relevant *in vitro* model by co-treating confluent ARPE-19 cells with different concentrations of TGFB2 and TNF-α, hereafter referred to as 'TNT'.

Nicotinamide (NAM), also known as niacinamide or vitamin B3, is a water-soluble derivative of vitamin B3 and a precursor of nicotinamide adenine dinucleotide (NAD⁺), a key cofactor in cellular redox reactions, DNA repair, and stress signaling. NAM

has demonstrated anti-inflammatory, antioxidative, and anti-fibrotic properties in a range of disease models, including cancer, neurodegeneration, and tissue fibrosis [14,15]. Mechanistically, NAM is thought to act through the modulation of $NAD^+$-dependent pathways and inhibition of SIRT1, thereby influencing gene expression and cellular stress responses involved in EMT progression [16]. Given its reported ability to inhibit TGFB2-induced epithelial-to-mesenchymal transition (EMT) and fibrosis in various cell types, we investigated whether NAM could similarly counteract the effects of combined TNF and TGFB2 (TNT) treatment in ARPE-19 cells.

In this study, we aimed to establish and characterize a TNT-based, high-throughput ARPE-19 live-cell imaging assay that recapitulates early retinal detachment–associated EMT and fibrotic remodeling. Using nicotinamide (NAM) as a proof-of-concept compound, we further show that reductions in TNT-driven nuclear migration allow early identification, within the first 10 h of live-cell imaging, of substances that prevent the formation of dense, multilayered contractile RPE aggregates, without the need for end-point analysis.

## Materials and methods

### Cell culture and cytokines treatments

The ARPE-19 human RPE cell line (ATTC #CRL-2302) was cultured in DMEM/F12 medium (Gibco #21331−020) supplemented with 1 x GlutaMAX™ (Gibco #3505−061), 5% FBS (Pan-Biotech #P30-3606), and 2% Penicillin-Streptomycin (Gibco #15140−122).

For cytokine treatments, $1 \times 10^5$ ARPE-19 cells were seeded in 24-well plates with 0.5 mL of complete medium containing 5% FBS. After 24 hours, the medium was replaced with fresh medium containing 2% FBS, 28 nM Hoechst (Sigma # B2261-25MG), and one of the following conditions (unless stated otherwise): 10 ng/mL TGFB2 (ThermoFisher #100-35B-10UG), 5 ng/mL TNF (ThermoFisher #300-01-10UG), 20 mM Nicotinamide (Sigma-Aldrich #N0636-100G), or their combinations. Cells were cultured for up to six days, with medium replenished every 72 hours.

### Live-cell imaging

Live-cell imaging (LCI) was performed using the BioTek® Lionheart™ FX automated microscope. Imaging settings were applied as previously described, with minor modifications [17]. After cytokine treatment, the 24-well plate was placed into the humidity chamber maintained at 37 °C and 5% $CO_2$. Imaging was conducted using the phase-contrast channel and a 4×PL FL objective. Hoechst (28 nM) was directly added to the culture medium to achieve nuclear staining. Laser autofocus was performed in the phase-contrast channel to determine the focal plane, which was then used for imaging Hoechst-stained nuclei via the DAPI filter, to minimize prolonged blue light exposure [18]. To minimize further phototoxicity due to blue light exposure, the 'LED intensity' was set to level 6, with an 'acquisition time' of 70 ms, and the maximum gain of 24 was used. Default autofocus and autoexposure settings were applied in the phase-contrast channel, and images were acquired every 20 minutes over a 72–144-hour period.

### Cell tracking

Cell migration was analyzed using the TrackMate plugin in Fiji (ImageJ) [19]. Time-lapse DAPI image sequences were first converted into 8-bit grayscale. The 'Set scale' tool was used to convert pixels into micrometers (μm), and, under 'Image Properties' the time interval was set to 20 minutes to reflect the acquisition rate.

For migration tracking, the Fiji Plugin 'TrackMate' was used. First, the nuclei in each image were detected, using the 'StarDist' detector plugin [20,21]. For tracking, the LAP tracker was applied, setting 'Frame to Frame Linking' to 20 μm, 'Track segment splitting' to 15 μm, and 'Track segment merging' to 15 μm. Finally, Cell migration trajectories were visualized with a temporal projection of 10 hours 'backwards in time' and color-coded according to instantaneous velocity (μm/min). The resulting videos were converted to MP4 video files.

 

## ICC staining

For ICC staining, $1 \times 10^5$ ARPE-19 cells were seeded on a 13 mm diameter slide coverslip (Glaswarenfabrik Karl Hecht #41001113) in each well of a 24-well plate with 0.5 mL complete medium. Cells were washed twice with PBS (Carl Roth #1058.1) before fixation with 4% paraformaldehyde (Carl Roth #P087.6) for 20 min at RT, followed by two additional 10 min washes in PBS. Blocking was conducted with a solution composed of 5% goat serum (Millipore #S26-100 ml), 0.02% Tween®20 (Sigma #P9416), and 0.01% Triton™X-100 (Sigma #X100) in PBS for 1 h at RT.

Cells were incubated overnight at 4°C with 1:1000 dilutions of Fibronectin 1 (FN1) rabbit mAb (Cell Signaling #26836S), COL1A1 rabbit mAb (Cell Signaling #72026S), or ZO-1 rabbit mAb (Cell Signaling #13663S) in blocking solution. After two washes with PBS for 10 min at RT, cells were incubated with AlexaFluor™488 goat anti-rabbit (Invitrogen #A11034, 1:2000) and Rhodamine-phalloidin (Invitrogen #R415, 4 µL/mL) in PBST (0.1% Tween®20 in PBS) for 2 h at RT. After three washes in PBS for 10 min at RT, the coverslips were mounted upside down on a slide with Roti®-Mount FluorCare DAPI (Carl Roth #HP20.1). Phase-contrast and fluorescence images were recorded with an Observer Z.1 microscope (Carl Zeiss) using the ZEN-Blue analysis software (Carl Zeiss). Representative images were chosen from three biological replicates.

## Western blot

$4 \times 10^5$ ARPE-19 cells were cultured in 6-well plates with 2 mL of complete medium. After 24h, cells were treated with TGFB2, TNF, TNT, or control with or without NAM for five days before lysis. Cells were lysed in $1 \times$ Laemmli Sample Buffer (Bio-Rad #1610737) supplemented with a $1 \times$ protease inhibitor cocktail (Cell Signaling #5871S). The lysates were mixed with 2-mercaptoethanol (Sigma-Aldrich #60-24-2), denatured at 95°C for 5 min, and stored at −20°C. Protein samples were loaded onto TGX Stain-Free™ Fast-Cast™ polyacrylamide gels (Bio-Rad #1610181) and separated by electrophoresis at 80–120 V. Resolved proteins were transferred onto an ethanol-activated Mini-size LF PVDF membrane (Bio-Rad #10026934) using the Trans-Blot® Turbo™ Transfer System at 1.3 A, 25 V for 7 min, or 10 min for high-molecular-weight proteins. After transfer, total protein bands were visualized using the 'Stain-Free blot' option of the ChemiDoc™ Imaging System following 45 sec UV activation and automatic exposure acquisition. Membranes were blocked with 5% milk powder (Carl Roth #T145.2) in $1 \times$ Tris-buffered saline (TBS) at room temperature (RT) for 1 h, followed by overnight incubation at 4°C with a 1:1000 dilution of Fibronectin 1 (FN1) rabbit mAb (Cell Signaling #26836S) and COL1A1 rabbit mAb (Cell Signaling #72026S). After two 5-minute washes in TBS, membranes were incubated with a 1:1000 dilution of goat anti-rabbit secondary antibody StarBright™ Blue 700 (Bio-Rad #12004162) at RT for one hour. Fluorescence signals were detected using the ChemiDoc™ MP Imaging System (Bio-Rad) at an excitation/emission wavelength of 660–720 nm. Image quantification and normalization were performed using ImageLab 6 software (Bio-Rad). Protein band intensities were normalized to the corresponding total protein signal instead of a housekeeping protein. The calculated densitometric ratio was performed from three to four biological replicates. Raw western blots detecting FN1 or Col1A1, and their corresponding total-protein blots used for normalization, are shown in S1, S2, S6 and S7 Figs.

## ELISA assays

To assess VEGFA secretion, conditioned media from ARPE-19 cells were collected at 24, 48, and 72 hours and stored at −80 °C until analysis. Before measurement, samples were thawed and centrifuged at $14,000 \times g$ for 5 minutes, and $3 \times 100$ µL aliquots of each sample were used for quantification. VEGFA levels were determined using the Human VEGF-165 Development Kit (TMB) (Peprotech #99-T10K) according to the manufacturer's instructions. Absorbance was measured using a Tecan Spark® M10 plate reader. Each condition was analyzed in three technical replicates across three biological replicates (n = 3).

## Gel contraction assay

Collagen gels were prepared according to the manufacturer's instructions (Corning #354249), yielding a final collagen concentration of 2.5 mg/mL. A volume of 500 μL of the collagen mixture was added to each well of a 24-well plate and incubated at 37 °C for 30 minutes to allow gel polymerization. After gelation, $1 \times 10^5$ ARPE-19 cells were seeded on top of each gel and cultured under standard conditions. After 24 hours, the medium was replaced with the cytokine-containing medium as described above. To initiate contraction, the gels were carefully detached from the well walls using a 200 μL pipette tip. The gel area was recorded daily over five days, and gel contraction was quantified indirectly by changes in the gel area using Fiji (ImageJ). For statistical analysis, three to five biological replicates were analyzed.

## Data analysis and statistics

Statistical analysis was performed using ANOVA in GraphPad Prism (Version 9.2.0). Migration parameters were extracted via TrackMate and exported for further processing. Western blot band intensities were quantified using ImageLab 6 (Bio-Rad). A p-value of $p < 0.05$ (*), $p < 0.01$ (**), $p < 0.001$ (***), and $p < 0.0001$ (****) was considered statistically significant.

## Results

### TGFB2 and TNF have a synergistic effect on EMT

In our previous studies, we characterized the effects of TGFB2 on ARPE-19 cells and primary human RPEs, demonstrating its role in promoting EMT-associated changes [17,22]. However, the vitreous humor contains a complex mixture of cytokines rather than isolated factors. Notably, cytokine profiling of vitreous humor samples has revealed the simultaneous presence of pro-fibrotic mediators such as TGFB2 and pro-inflammatory cytokines like TNF [23,24]. To better mimic this physiological context and explore potential synergistic effects, we investigated the combined influence of TGFB2 and TNF on EMT progression in ARPE-19 cells. Therefore, ARPE-19 cells were assigned to four experimental groups: control, TGFB2, TNF, and a combination of TGFB2 and TNF, hereafter referred to as 'TNT'.

Phase-contrast imaging at 72 hours post-treatment revealed distinct morphological alterations. Compared to the other three groups, TNT-exposed cells exhibited pronounced elongation and a distinct directional alignment (Fig 1A). To further characterize these changes, immunocytochemistry (ICC) was performed to assess the expression and spatial distribution of the mesenchymal extracellular matrix proteins Fibronectin 1 (FN1) and Alpha-1 Type I Collagen (COL1A1). Phalloidin staining was used to visualize actin filament organization. COL1A1 expression in the TGFB2 and TNF groups closely resembled that of the control group, with no significant alterations in localization or intensity. In contrast, the TNT group displayed a strong COL1A1 signal with localized ECM aggregation. FN1 showed punctate deposition in the TNF group and more pronounced patchy aggregates in the TGFB2 group. In the TNT group, FN1 formed extensive network-like deposits, suggesting a more advanced stage of ECM remodelling. While F-actin fibers in the control group were radially organized around the nucleus, the TNF group exhibited a more disorganized structure. In the TGFB2 group, actin filaments appeared denser with some degree of parallel alignment. Remarkably, TNT-treated cells showed a filament network similar to the TGFB2 group but with more pronounced elongation and directional organization (Fig 1B).

To quantify protein expression, Western blot analysis was performed for FN1 and COL1A1. FN1 levels were significantly elevated in both TGFB2- and TNF-treated cells, with slightly higher expression observed in the TGFB2 group (Fig 1C, D). The TNT group exhibited the highest FN1 expression, significantly surpassing all other groups. Similarly, COL1A1 protein levels were markedly increased in the TNT group. While slight elevations were also observed in the TGFB2 and TNF groups, these did not reach statistical significance when compared to the control.

 

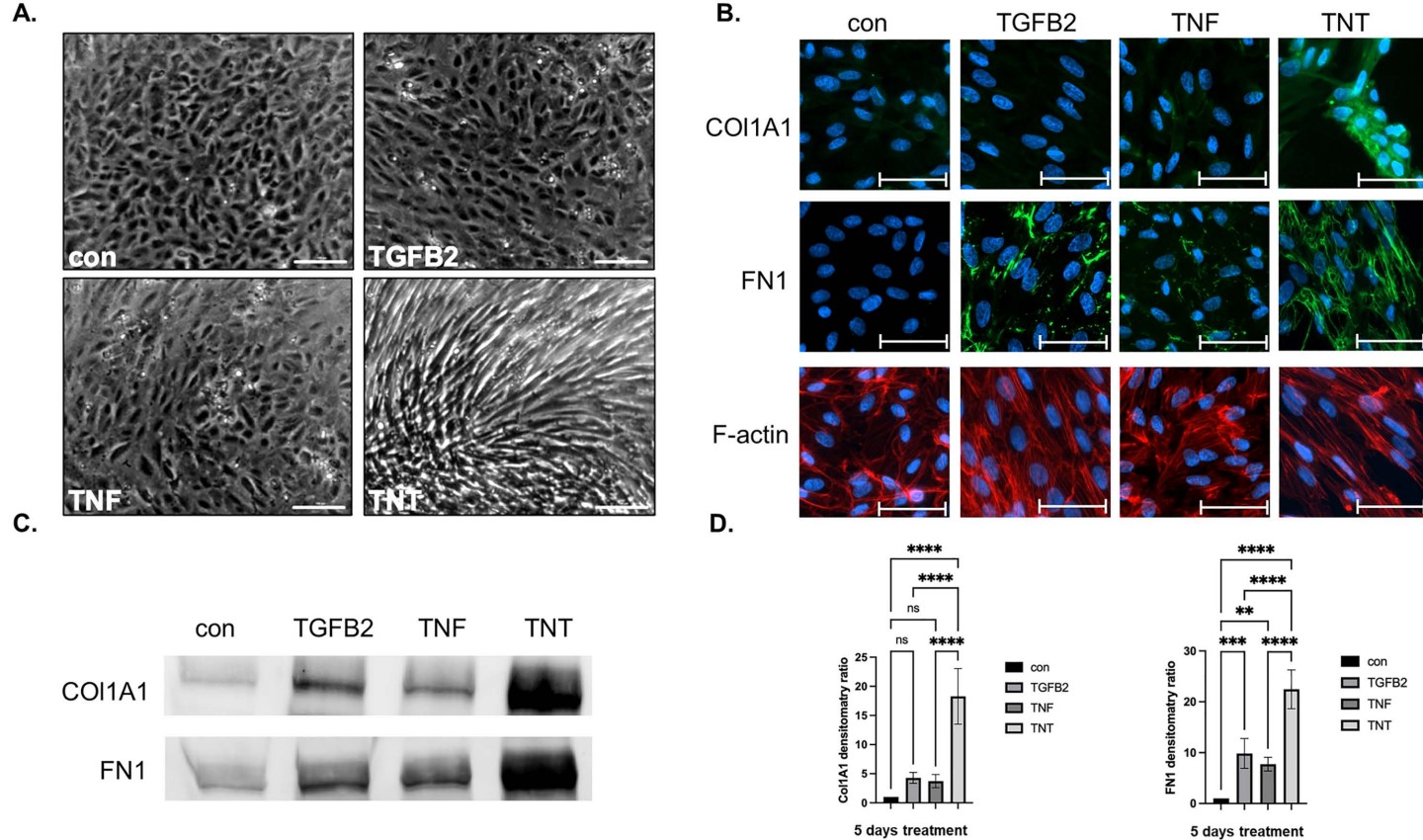

**Fig 1. TGFB2 and TNF have a synergistic effect on Fibrosis. (A)** The morphology of ARPE-19 cells after 3 days of treatment in control, TGFB2, TNF, and TGFB2 + TNF (TNT) exposed groups. The scale bar represents 500 μm. **(B)** Three days after treatment, cells were immunostained for COL1A1, FN1, and F-actin. The scale bar represents 50 μm. **(C)** Representative Western blot of FN1 and COL1A1 in ARPE-19 cells after five days of treatment. **(D)** The densitometric ratio of FN1 and COL1A1 protein was normalized to total proteins. The values were normalized to the control group. The raw western blot images used for quantification are shown in Supplementary Information (S1 and S2 Figs). One-way ANOVA with Tukey's multiple comparisons test (n = 4) was performed. The error bars represent SD. Ns, non-significant, **p < 0.01, ***p < 0.001, ****p < 0.0001.

## TNT remarkably enhances the mobility of ARPE-19 cells and promotes the formation of multi-layered cell clusters

To further explore the impact of the treatments on cellular dynamics, we next conducted live-cell imaging combined with Hoechst staining to monitor migration patterns and multicellular behavior in real-time. Cells in the TGFB2 group exhibited elongation and partial alignment compared to the control, 72 hours after cytokine exposure (Fig 2A). In the TNF group, elongation was more pronounced, accompanied by the formation of multicellular aggregates. Notably, the TNT group displayed not only multicellular aggregates but also substantial cellular aggregation, occasionally extending beyond 100 μm, as revealed by nuclear staining.

To track nuclear migration, we used the FIJI plugin TrackMate in combination with the StarDist detector, which enabled robust and automated tracking of Hoechst-stained nuclei in LCI sequences over 10-h time windows preceding each time point. Our analysis revealed that TNT-treated cells exhibited rapid migration along well-defined trajectories, a behavior not observed in the other three groups (S2 and S3 Videos). Interestingly, the convergence of these cells at defined aggregation sites reflects a pattern characteristic of coordinated group migration. These results confirm that the TNT group exhibited significantly enhanced migratory activity compared to the other conditions (Fig 2B). For visual clarity, nuclei were

**A.**

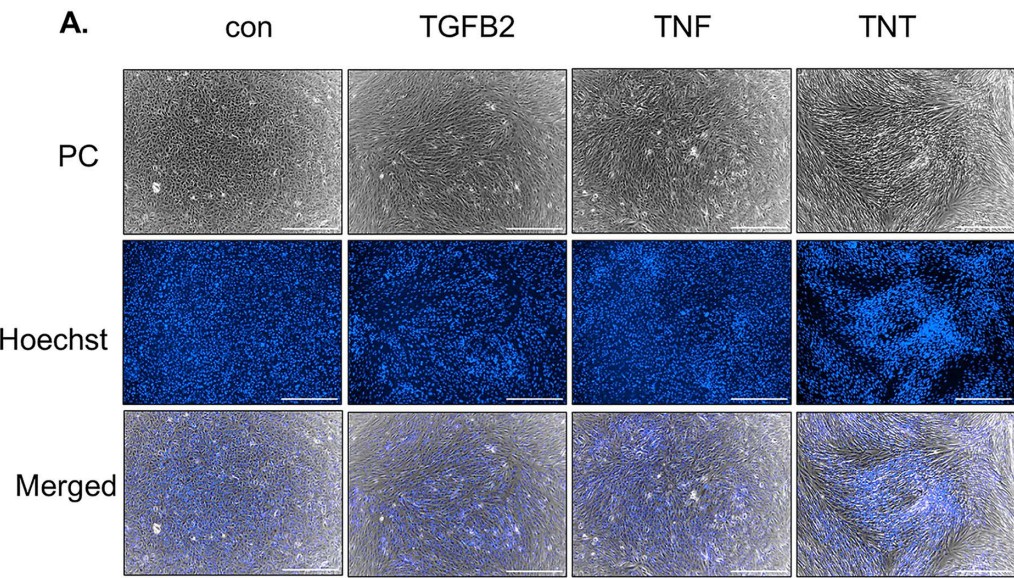

**B.**

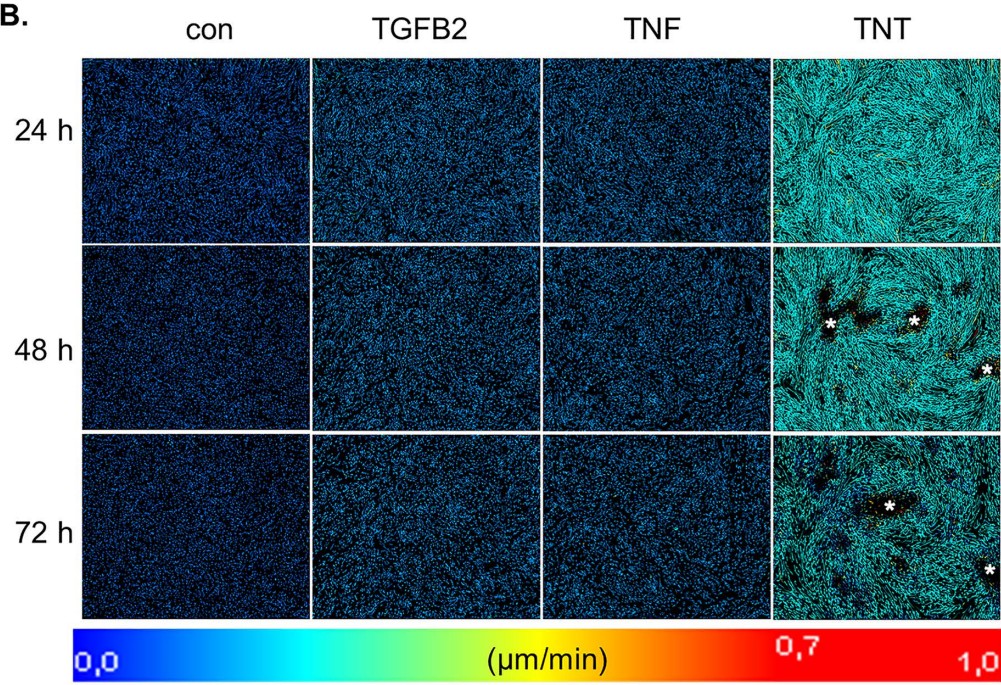

**Fig 2. TNT enhances the mobility of ARPE-19 cells and promotes the formation of multi-layered cell clusters. (A)** Phase contrast images, fluorescence images of Hoechst 33342-stained nuclei, and merged images of ARPE-19 cells after three days of treatment with TGFB2, TNF, or TNT are shown. The scale bar represents 500 μm. **(B)** Representative nuclear trajectory and velocity maps of ARPE-19 cells under the indicated treatment conditions at 24, 48, and 72 hours. Trajectories were reconstructed over the 10 h preceding each time point using the TrackMate/StarDist pipeline, and track colour encodes instantaneous nuclear velocity (μm/min) according to the scale bar. Apparent black regions within areas of high track density (marked by asterisks) correspond to densely packed nuclei in multicellular aggregates, which could no longer be segmented individually by the detection algorithm. The correspondence between these regions and Hoechst-positive multicellular aggregates is illustrated in S3 Fig, S2 and S3 Videos.

hidden in the main trajectory maps so that tracking paths remained unobstructed. In TNT-treated wells, nuclei frequently formed dense, multilayered aggregates, at which point TrackMate could no longer reliably segment individual nuclei, resulting in apparent 'gaps' in the track maps. To make this explicit, we include trajectories reconstructed over the 10 h preceding the 0 h, 24 h, 48 h, and 72 h time points together with the nuclei overlay from a representative TNT experiment (S3 Fig).

Next, we examined how varying TNF concentrations (0, 1, 5, and 10 ng/mL) at a constant TGFB2 concentration of 10 ng/mL influence the formation of multicellular aggregates (S4 Fig). The 4 × 4 phase-contrast image montages, acquired three days post-exposure using a 4 × objective for improved overview, reveal a dose-dependent modulation of aggregate formation, such that the severity of the phenotype can be tuned according to the experimental objective.

To determine whether these differences in aggregate formation are preceded by changes in cell motility, we performed a separate tracking experiment in which confluent ARPE-19 cells were exposed to different ranges of TNF concentrations (0, 1, 5, and 10 ng/mL) in the presence of 10 ng/mL TGFB2. Migration trajectories and velocities were retrospectively analyzed over the 10 h preceding the 24 h, 48 h, and 72 h time points (S5 Fig). The tracking data indicate that, within the analysed 10 h windows, nuclear trajectories are faster and more persistent at higher TNF concentrations, consistent with enhanced collective migration before any detectable detachment.

## TNT significantly enhances the contraction ability of ARPE-19 cells

LCI over five days of TNT treatment revealed pronounced morphological changes indicative of an EMT phenotype. In some instances, cells detached from the well bottom following TNT exposure, as documented in the Supplementary Video (S1 Video). A Collagen I-based GCA was performed to quantitatively assess their associated contractile properties.

After 72 hours of treatment, the gel area in the TGFB2 group showed a slight reduction compared to the control, while the TNF group exhibited a more pronounced decrease. Notably, the TNT group displayed the strongest contraction, with all differences reaching statistical significance (Fig 3A, B). To assess contraction dynamics, gel areas were monitored over five days, revealing distinct temporal patterns among the groups.

In the control group, contraction progressed at a relatively constant rate. In contrast, all three treatment groups exhibited the most pronounced gel shrinkage between day one and day two, followed by only minor changes from day four onward (Fig 3C).

## Nicotinamide suppresses the TNT-induced EMT process in ARPE-19 cells

Next, we analyzed the impact of NAM on cell morphology, EMT marker expression, and TNT-induced migration and formation of multicellular aggregates. Phase-contrast microscopy and Hoechst staining revealed that TNT-treated cells exhibited pronounced elongation, multilayered structures, and aggregation, indicative of EMT progression. In contrast, NAM co-treatment attenuated cell elongation and multilayering, resulting in a morphology comparable to the control group. Live-cell imaging further demonstrated that, while TNT-treated cells underwent significant contraction and detachment over five days, NAM co-treatment preserved adhesion and minimized contraction (Fig 4A).

ICC revealed a marked increase in FN and COL1A1 deposition in the TNT-treated group, characterized by dense, fibrous alignment within the ECM. Co-treatment with NAM significantly reduced the fluorescence intensity of both proteins and disrupted their alignment, indicating an inhibitory effect on ECM remodeling (Fig 4B). These findings were corroborated by WB analysis, which confirmed that FN1 and COL1A1 expression levels peaked under TNT treatment but were significantly reduced with NAM co-treatment (Fig 4C, D). Consistent with these ECM changes, ZO-1 immunostaining after 5 days showed that TGFB2 and, more prominently, TNF and TNT disrupted the continuous junctional ZO-1 belt and induced a more cytoplasmic staining pattern. NAM co-treatment preserved a near-cobblestone, membrane-associated ZO-1 distribution in control and TGFB2 groups and partially restored junctional ZO-1 under TNF. Under TNT conditions, junctional ZO-1 was largely lost and only faintly detectable, whereas discontinuous membrane staining reappeared in the TNT + NAM group (S8 Fig).

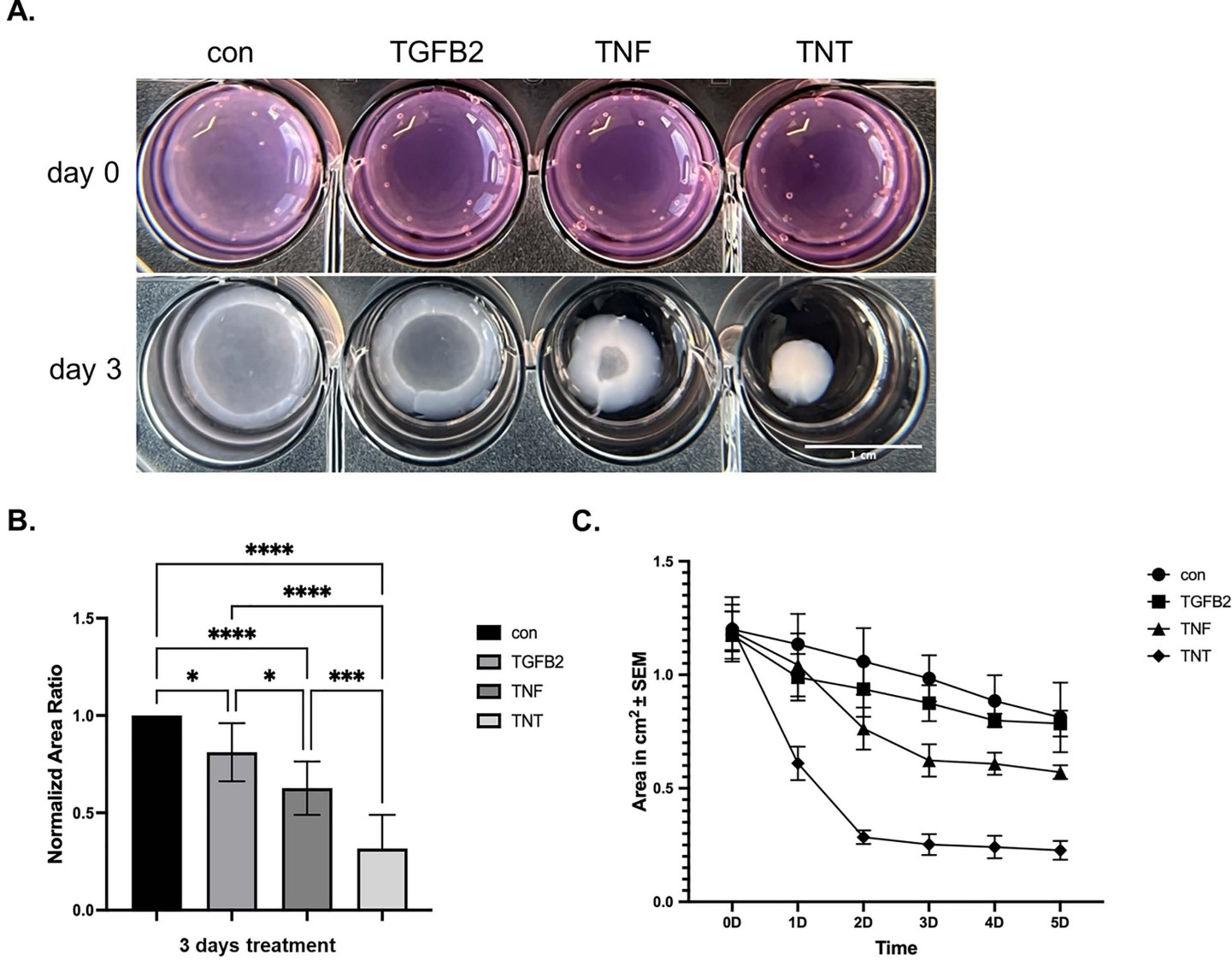

**Fig 3. TNT significantly enhances the contraction ability of ARPE-19 cells. (A)** The Images of collagen gel contraction under different treatments (control, TGFB2, TNF, and TNT) over three days. The scale bar represents 1 cm. **(B)** The normalized area ratio of collagen I gel contraction after three days of treatment. The values were normalized to the control group. One-way ANOVA with Tukey's multiple comparisons test (n = 3) was performed. The error bars represent SD. ns, non-significant, **$p < 0.01$, ***$p < 0.001$, ****$p < 0.0001$ **(C)** This collagen gel contraction area (cm² ± SEM) over 5 days for different treatment groups.

Our LCI tracking analysis further revealed that NAM reduced the migration speed of both control and TNT-treated cells over 72 hours, while cells exposed to TNT alone retained the rapid, directional migration culminating in aggregation, as previously described (Fig 5A, S4 Video).

In line with these observations, GCA assays showed that TNT-treated cells exhibited the strongest contractile activity, whereas NAM-exposed TNT-treated cells showed reduced contraction dynamics resulting in a significantly reduced gel shrinkage (Fig 5B, C).

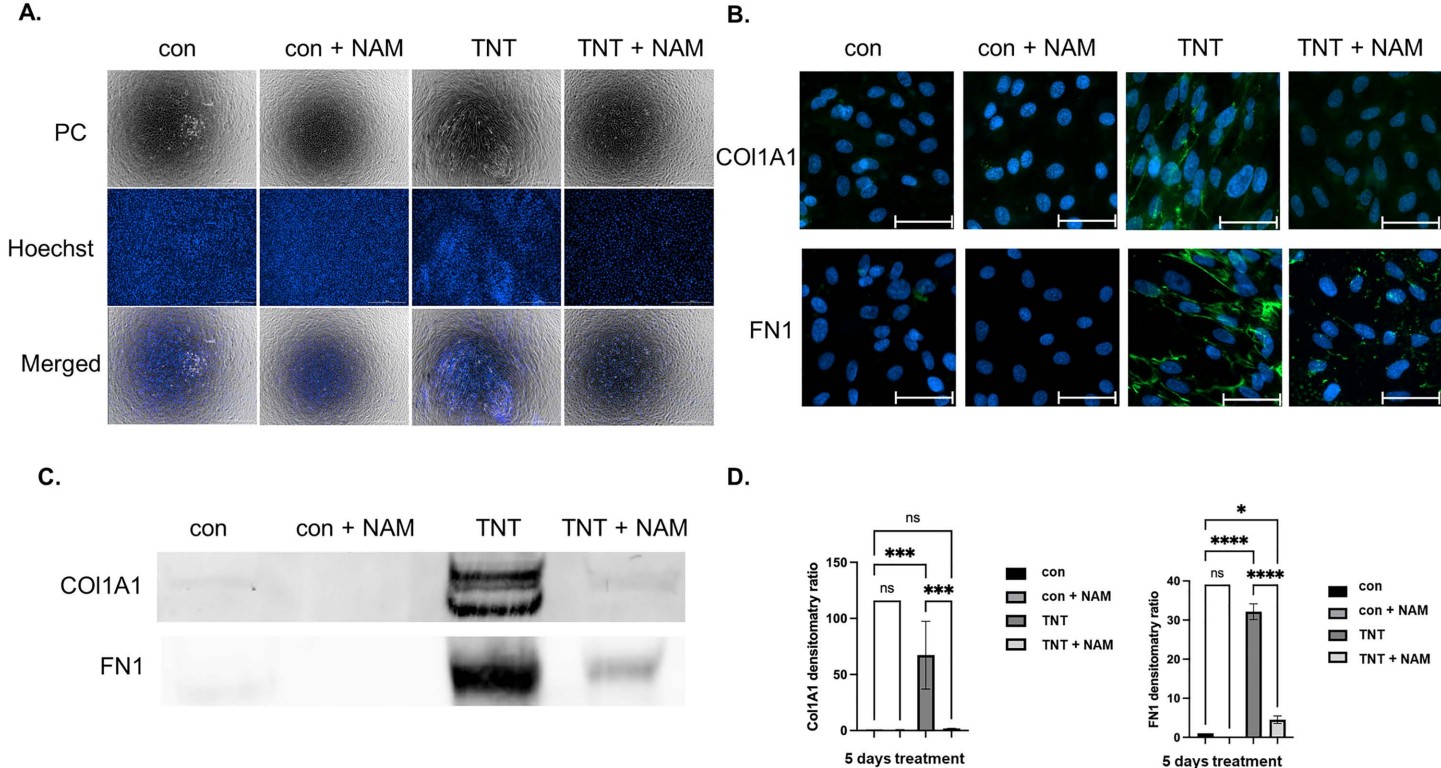

**Fig 4. Nicotinamide suppresses TNT-induced EMT in ARPE-19 cells. (A)** The morphology of ARPE-19 cells and nuclear staining after three days of treatment with control, control + NAM, TNT, and TNT + NAM. The scale bar represents 500 μm. **(B)** Cells were immunostained for COL1A1 and FN1 after 3 days of treatment. The scale bar represents 50 μm. **(C)** Representative Western blot of FN1 and COL1A1 in ARPE-19 cells after 5 days of treatment. **(D)** The densitometric ratio of FN1 and COL1A1 protein was normalized to total proteins. The values were normalized to the control group. One-way ANOVA with Tukey's multiple comparisons tests (n = 3) was performed. The error bars represent SD. ns, non-significant, *p < 0.05, ***p < 0.001, ****p < 0.0001. The raw Western blot images used for quantification are shown in Supplementary Information (S6 and S7 Figs).

## Nicotinamide suppresses TNT-induced VEGFA secretion in ARPE-19 cells

Since EMT is not only associated with fibrotic remodeling but also with enhanced secretion of pro-angiogenic factors such as VEGFA, and given that VEGFA is a key target of several translational strategies in ophthalmology, we sought to determine whether TNT treatment modulates VEGFA secretion in ARPE-19 cells. VEGFA secretion levels were quantified via ELISA assays in the culture medium collected at 24, 48, and 72 hours post-treatment (Fig 6).

Compared to the control group, VEGFA secretion was significantly increased at 24 h, 48 h, and 72 h in both the TGFB2 and TNT groups, whereas a significant elevation was observed in the TNF group only after 72 h. Additionally, we investigated the effect of NAM on TNT-induced VEGFA secretion in ARPE-19 cells. The results showed that VEGFA secretion at all three time points was significantly lower in the TNT + NAM group compared to the TNT group alone, with all differences reaching highly statistical significance.

## Discussion

PVR is a complex, multifactorial fibrotic retinal disorder characterized by epiretinal/subretinal formation of fibrotic membranes, primarily driven by EMT of RPE cells. While EMT of RPE cells plays a central role, PVR progression involves multiple contributing factors, including inflammation, proliferation, migration of diverse cell populations, and alterations in

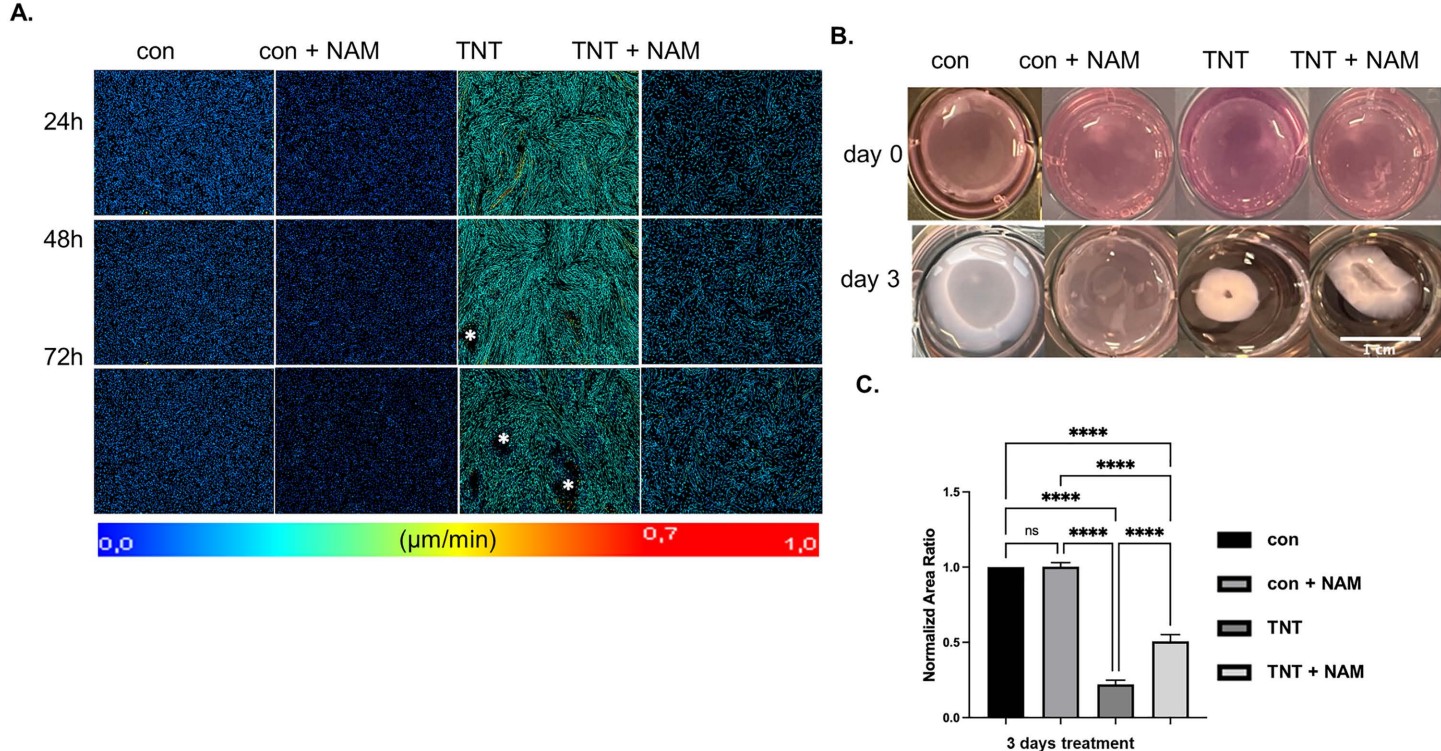

**Fig 5. NAM inhibited TNT-induced migration and contractile activity in ARPE-19 cells. (A)** Representative cell migration velocity maps of ARPE-19 cells treated with control, control + NAM, TNT, and TNT + NAM for 24 hours. The color scale represents migration velocity, ranging from 0.0 to 1.0 μm/min. The regions (marked by white asterisks) represent cell aggregates. These black areas appear due to the failure of gating by the StarDist detection algorithm. **(B)** Representative images of collagen gel contraction under different treatments after three days. The scale bar represents 1 cm. **(C)** The normalized area ratio of collagen I gel contraction after three days of treatment. One-way ANOVA with Tukey's multiple comparisons tests (n = 5) was performed. The error bars represent SD. ns, non-significant, ****p < 0.0001.

extracellular matrix remodeling. During PVR, RPE cells lose epithelial polarity and adhesion, gain contractile and migratory capabilities, and actively participate in fibrotic membrane formation. These transformations are orchestrated by a complex interplay of cytokines and growth factors present within the ocular microenvironment, including interleukin-6 (IL-6), interleukin-8 (IL-8), tumor necrosis factor-α (TNF-α), monocyte chemoattractant protein-1 (MCP-1), placental growth factor (PIGF), vascular endothelial growth factor (VEGF), transforming growth factor-β1 (TGF-β1), and transforming growth factor-β2 (TGF-β2) [24–27].

However, many *in vitro* models rely on single-cytokine stimulation, which may insufficiently capture the multifactorial nature of PVR pathogenesis. Recent work has begun to address this gap: in adult primary human RPE (ahRPE), co-stimulation with TGFB1 and TNF synergistically induces EMT, aggregate formation, and membrane contractility via p38 signaling, recapitulating key PVR features *in vitro* [28]. Similarly, Boles et al. showed that TNT treatment enhances EMT primary human RPE cells as well as aggregate formation and identified core regulatory factors through epigenomic and transcriptomic analyses [29]. Nonetheless, while primary hRPE cells are physiologically relevant, they pose challenges in terms of accessibility and experimental reproducibility.

To address this limitation, we established a co-treatment model using TNT in ARPE-19 cells, a well-characterized, readily available human RPE cell line. Although ARPE-19 cells differ phenotypically from primary or iPSC-derived RPE, they offer advantages in terms of reproducibility, culture stability, and scalability for mechanistic and screening studies.

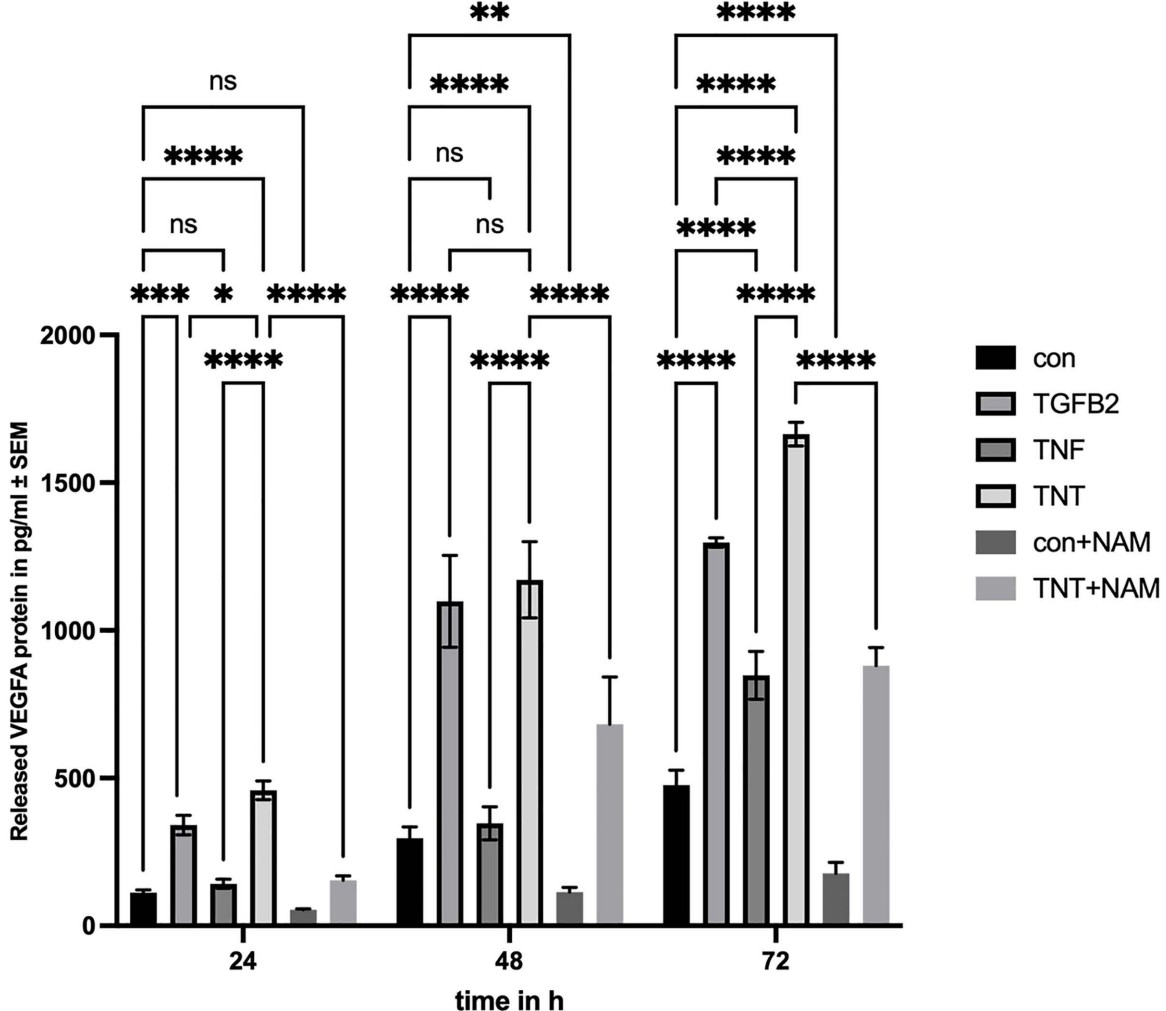

**Fig 6. Nicotinamide suppresses TNT-induced VEGFA secretion in ARPE-19 cells.** Released VEGFA protein (pg/mL) in different groups after 24, 48, and 72 h of treatment was quantified by ELISA. Two-way ANOVA with Tukey's multiple comparisons test (n = 3) was performed. The error bars represent SD. ns, non-significant, *$p < 0.05$ **$p < 0.01$, ***$p < 0.001$, ****$p < 0.0001$.

However, as an immortalized and partially dedifferentiated cell line, ARPE-19 cells may exhibit an EMT-like baseline phenotype and altered cytokine responsiveness compared with native or iPSC-derived RPE, which could lead to an overestimation of the absolute magnitude of EMT/fibrotic responses and inhibitory effects observed in our model. While models based on primary or iPSC-derived RPE or more complex cytokine environments may better recapitulate the *in vivo* pathophysiology of PVR, they are often experimentally demanding and less suited for high-throughput approaches. The TNT-ARPE-19 system represents a practical and informative platform for dissecting EMT dynamics, particularly in early-stage exploratory research. Notably, consistent with prior reports in primary human RPE cells [28,29], we also observed robust multicellular aggregate formation under TNT co-treatment in ARPE-19 cells, indicating that this aggregation phenotype is reproducible and can be translated from primary hRPE to this widely used cell line.

For real-time monitoring, we used live-cell imaging and automated cell tracking, integrating Hoechst nuclear staining with TrackMate (Fiji) [18,19]. This approach was designed to elucidate how directed migration and local cell–cell interactions give rise to multicellular aggregate formation. This LCI approach further enabled high-resolution visualization of

dynamic cellular changes, including morphological alterations, as well as contraction and rapture events often observed in the TNT-treated group (S1 Video). TrackMate analysis using Fiji (Formerly ImageJ) further revealed that TNT treatment significantly enhanced directed cell motility. Notably, a marked increase in migration speed was already detectable at 24 hours in the TNT group, and comparing the TNT trajectories with those under NAM + TNT at this early time point allowed us to infer impending aggregate formation (trajectory convergence and local densification of tracks in TNT). A directional migration pattern resembling collective cell migration was observed, characterized by coordinated movement and alignment of neighboring cells, ultimately leading to the formation of multicellular cell aggregates. In addition, collagen contraction assays revealed a significant increase in contractile activity following TNT treatment, reinforcing the functional transition associated with EMT in this model. Thus, within the first 24 hours, LCI allows to infer whether a test compound attenuates TNT-induced aggregate formation, without the need to await endpoint assays.

Importantly, we observed a concentration-dependent effect of TNF, with lower levels resulting in reduced motility and cluster formation (S3 and S4 Figs). This finding underscores the modulatory role of inflammatory cytokines in EMT intensity and suggests that TNF may act as a para-inflammatory amplifier of fibrosis in the ocular environment. Such dose-sensitive effects have implications for understanding PVR severity and for the development of stage-adapted therapeutic interventions.

In addition to mechanistic insights, our study explored the therapeutic potential of nicotinamide (NAM), a vitamin B3 derivative with known anti-fibrotic and anti-inflammatory effects [30–35]. While NAM has previously been shown to inhibit EMT in adult hRPE by modulating transcriptional regulators [29], its role in other RPE models, such as ARPE-19 cells, has not been well defined.

In ARPE-19 cells, we show that NAM significantly suppressed TNT-induced EMT features, including cell elongation, ECM deposition, migration, contractility, and importantly, VEGFA secretion. The latter is particularly relevant given the dual role of EMT in promoting both fibrosis and pathological angiogenesis in retinal diseases.

In summary, our study establishes the TNT-ARPE-19 system as a robust and pathophysiologically relevant *in vitro* model that recapitulates key fibrotic and pro-angiogenic features of proliferative vitreoretinopathy. Co-stimulation with TGFB2 and TNF synergistically induced EMT-like phenotypes, including cytoskeletal remodeling, extracellular matrix deposition, increased contractility, multicellular aggregation, and elevated VEGFA secretion. Notably, the model allows real-time tracking of cell migration and phenotypic transitions using live-cell imaging and automated nuclear tracking, enabling early identification of EMT dynamics without reliance on endpoint-based assays. The fibrotic phenotype can be modulated by adjusting TNF concentrations, allowing for the simulation of mild to aggressive disease trajectories.

NAM significantly attenuated both fibrotic and angiogenic responses, supporting its potential as a modulator of EMT in retinal pigment epithelial cells. While the model does not replicate the full complexity of the vitreous microenvironment, it captures essential aspects of PVR pathomechanisms and provides a reproducible and scalable platform for mechanistic studies and compound screening under defined conditions. The use of the immortalized ARPE-19 cell line ensures consistency across experiments and facilitates high-throughput approaches. A further limitation of the present study is that we did not perform a dedicated, systematic viability assay under all treatment conditions, although initial propidium iodide staining did not reveal overt TNT-induced cell death. Future refinements may include the addition of further cytokines such as IL6, IL8, or IFNG to better approximate inflammatory profiles associated with different PVR subtypes. Moreover, validation of the observed effects in primary RPE cells or more complex co-culture and 3D models could enhance the physiological relevance and inform translational applications.

## Supporting information

**S1 Fig. Uncropped Western blot membranes used for COL1A1 detection in Fig 1C.** (A) RAW Western blot image showing COL1A1 protein used for densitometric quantification presented in Fig 1C. Each lane represents protein lysates from ARPE-19 cells treated under the indicated conditions. (B) RAW Western blot image showing total protein bands used

for densitometric quantification presented in Fig 1C. Each lane represents protein lysates from ARPE-19 cells treated under the indicated conditions.

(PDF)

**S2 Fig. Uncropped Western blot membranes used for FN1 detection in Fig 1C.** (A) RAW Western blot image showing FN1 protein used for densitometric quantification presented in Fig 4C. Each lane represents protein lysates from ARPE-19 cells treated under the indicated conditions. (B) RAW Western blot image showing total protein bands used for densitometric quantification presented in Fig 4C. Each lane represents protein lysates from ARPE-19 cells treated under the indicated conditions.

(PDF)

**S3 Fig. Migration velocity maps of ARPE-19 cells under TNT stimulation.** Color-coded migration velocity maps of ARPE-19 cells exposed to TNT for 24, 48, and 72 hours. Nuclear trajectories were tracked 10 h backwards in time using live-cell imaging and analyzed with the StarDist/TrackMate plugin in FIJI. Hoechst-stained nuclei are shown in grey. The scale bar represents 1000 µm.

(PDF)

**S4 Fig. Phase-contrast images of ARPE-19 cells after cytokine treatment.** Representative phase-contrast images of ARPE-19 cells three days after treatment with 10 ng/mL TGFB2 and varying concentrations of TNT (0, 10, 5, and 1 ng/mL TNF in combination with 10 ng/mL TGFB2). The scale bar represents 2000 µm.

(PDF)

**S5 Fig. Velocity-coded nuclear migration trajectories of ARPE-19 cells exposed to different TNF concentrations at a constant TGFB2 level.** Confluent ARPE-19 monolayers were treated with control medium (Con), 10 ng/mL TGFB2 + 10 ng/mL TNF, 10 ng/mL TGFB2 + 5 ng/mL TNF, 10 ng/mL TGFB2 + 1 ng/mL TNF, or 10 ng/mL TGFB2 alone and subjected to live-cell imaging. Hoechst-stained nuclei were tracked with StarDist/TrackMat plugin in FIJI, and trajectories were reconstructed for a 10-h tracking window ending at the indicated 24 h, 48 h, and 72 h time points. Tracks are color-coded for instantaneous migration velocity according to the heat map (0–1.0 µm/min). Scale bar: 1000 µm.

(PDF)

**S6 Fig. Uncropped Western blot membranes used for COL1A1 detection in Fig 4C.** (A) RAW Western blot image showing COL1A1 protein used for densitometric quantification presented in Fig 4C. Each lane represents protein lysates from ARPE-19 cells treated under the indicated conditions. (B) RAW Western blot image showing total protein bands used for densitometric quantification presented in Fig 4C. Each lane represents protein lysates from ARPE-19 cells treated under the indicated conditions.

(PDF)

**S7 Fig. Uncropped Western blot membranes used for FN1 detection in Fig 4C.** (A) RAW Western blot image showing FN1 protein used for densitometric quantification presented in Fig 4C. Each lane represents protein lysates from ARPE-19 cells treated under the indicated conditions. (B) RAW Western blot image showing total protein bands used for densitometric quantification presented in Fig 4C. Each lane represents protein lysates from ARPE-19 cells treated under the indicated conditions.

(PDF)

**S8 Fig. ZO-1 immunocytochemistry after 5-day TGFB2/TNF ± NAM treatment.** ARPE-19 monolayers were exposed for 5 days to control medium (con), TGFB2, TNF, or the combination of TGFB2 + TNF (TNT) in the absence (left panel, "without NAM") or presence (right panel, "with NAM") of 20 mM nicotinamide (NAM). Cells were fixed and stained for ZO-1 (green) and counterstained with DAPI (blue); merged images are shown in the right column of each panel. TNT

corresponds to 10 ng/mL TGFB2 + 5 ng/mL TNF. Images are representative of three independent experiments. Scale bar: 50 μm.
(PDF)

**S1 Table. Raw data: All individual data points obtained from the respective experiments and used for the generation of Figs 1 D, 3 B, C, 4D, 5C, and 6A, B are provided.**
(XLSX)

**S1 Video. Phase-contrast live-cell imaging of control and TNT-treated ARPE-19 cells.** Phase-contrast live-cell imaging of control and TNT-treated ARPE-19 cells. After 72 h of pre-exposure to control medium or TNT (10 ng/mL TGFB2 + 5 ng/mL TNF), images were acquired every 20 minutes over an additional 72 h. TNT-treated cells display pronounced epithelial-mesenchymal transition (EMT)-related morphological changes and frequent cell detachment events, which are not observed in control cells. Scale bar: 1000 μm.
(MP4)

**S2 Video. Live-cell imaging of Hoechst-stained nuclei in control and TNT-treated ARPE-19 cells.** Following three days of pretreatment, nuclei were imaged every 20 minutes for 72 hours. Control cells show limited motility, whereas TNT-treated cells exhibit directional migration and aggregate formation. Scale bar: 1000 μm.
(MP4)

**S3 Video. Tracking of nuclear migration velocity in control and TNT-treated ARPE-19 cells.** Nuclear trajectories were detected using StarDist and tracked using the TrackMate plugin in Fiji. TNT-treated cells demonstrate significantly increased migration speed and directional movement toward aggregation points compared to controls. The color scale represents instantaneous velocity (0 to 1.0 μm/min). For clarity, nuclei are hidden in the visualization, and trajectories are shown retrospectively for the last 10 hours.
(MP4)

**S4 Video. Tracking of nuclear migration velocity in TNT-treated and TNT + NAM-treated ARPE-19 cells.** The upper row shows live-cell imaging (LCI) of Hoechst-stained nuclei used for tracking. The lower row displays the corresponding nuclear trajectories, detected using StarDist and analyzed via the TrackMate plugin in Fiji. TNT-treated cells exhibit significantly increased migration speed and directional movement toward aggregation points, while co-treatment with Nicotinamide (NAM) reduces both motility and aggregation. Scale bar: 1000 μm. The color scale represents instantaneous velocity (zero –1.0 μm/min).
(MP4)

## Author contributions

**Conceptualization:** Yuqing Huang, Heiko Fuchs.

**Data curation:** Yuqing Huang, Roland Meister, Heiko Fuchs.

**Formal analysis:** Yuqing Huang, Roland Meister, Heiko Fuchs.

**Investigation:** Yuqing Huang.

**Methodology:** Yuqing Huang, Heiko Fuchs.

**Project administration:** Heiko Fuchs.

**Resources:** Carsten Framme.

**Supervision:** Migle Lindziute, Maximilian Binter, Jan Tode, Heiko Fuchs.

**Visualization:** Yuqing Huang, Heiko Fuchs.

**Writing – original draft:** Yuqing Huang, Heiko Fuchs.

**Writing – review & editing:** Yuqing Huang, Migle Lindziute, Maximilian Binter, Heiko Fuchs.

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
