## [Decision Letter · Decision Letter 0]

31 Jul 2025

Dear Dr. Fuchs,

We look forward to receiving your revised manuscript.

Kind regards,

Andre van Wijnen, PhD

Academic Editor

PLOS ONE

Journal Requirements:

Reviewers' comments:

Reviewer's Responses to Questions

**Comments to the Author**

1. Is the manuscript technically sound, and do the data support the conclusions?

Reviewer #1: Yes

2. Has the statistical analysis been performed appropriately and rigorously?

Reviewer #1: Yes

3. Have the authors made all data underlying the findings in their manuscript fully available?

Reviewer #1: Yes

4. Is the manuscript presented in an intelligible fashion and written in standard English?

Reviewer #1: Yes

Reviewer #1: Dear author

In this manuscript, the authors investigated theTGFB2/TNF-induced in vitro model of Proliferative Vitreoretinopathy (PVR) using ARPE-19 cells. This study has generated some interesting data, however, this manuscript lacks some key data.

1.There is no internal control in the western blot shown in Figure 1 and 4.

2. Some data about mechanisms related to the induction of PVR by TGFB2/TNF are lacking in the text.

**Do you want your identity to be public for this peer review?** For information about this choice, including consent withdrawal, please see our Privacy Policy

Reviewer #1: **Yes: ** Xiaodong chen

---

## [Author Response · Author response to Decision Letter 1]

29 Aug 2025

Dear Prof. Xiaodong Chen,

We would like to thank you for the constructive feedback on our manuscript “A TGFB2/TNF-Induced in vitro model of Proliferative Vitreoretinopathy (PVR) using ARPE-19 cells confirms Nicotinamide as an inhibitor of EMT and VEGFA secretion”.

We carefully addressed all comments raised during the review process. Below, we provide a point-by-point response highlighting the changes implemented in the revised version of the manuscript.

Reviewer’s comment 1:

There is no internal control in the western blot shown in Figure 1 and 4.

Response:

Thank you for pointing this out. Instead of using a conventional loading control such as β-actin or GAPDH, in this study we normalized all western blot data to total protein levels, which were detected by stain-free imaging. This technique directly visualizes the total protein in each lane based on the intrinsic fluorescence of tryptophan residues, providing a reliable and unbiased reference for normalization. Representative stain-free images are now included for reference.

Unfortunately, this was not clearly described in the submitted version of the manuscript. We have therefore revised the Methods section to explicitly state:

“After transfer, total protein bands were visualized using the ‘Stain-Free blot’ option of the ChemiDoc™ Imaging System following 45 s UV activation and automatic exposure acquisition.”

“Protein band intensities were normalized to the corresponding total protein signal instead of a housekeeping protein.”

For reasons of space and figure clarity, these stain-free blots are not shown in the main figures but the corresponding raw stain-free total protein blots have been included in the Supplementary Figures SF3–SF6 to ensure full transparency.

In addition, as requested by the Editor, all raw data underlying the presented figures have been provided as a separate supplementary file Supplementary Table ST1 -raw data.xlsx.

Reviewer’s comment 2:

Some data about mechanisms related to the induction of PVR by TGFB2/TNF are lacking in the text.

Response:

We appreciate this insightful comment. In the present study, our main objective was to establish a reliable model of PVR induction by TGFβ2/TNF, and we therefore focused primarily on characterizing the phenotypic and functional changes in this model. We did not perform detailed mechanistic experiments at this stage. However, we have revised the Discussion to better contextualize our findings within known signaling mechanisms. Specifically, we now emphasize that TGFβ2 predominantly signals via Smad2/3 phosphorylation to drive EMT and extracellular matrix production, whereas TNF activates NF-κB and MAPK pathways to enhance inflammation and amplify fibrotic responses. Their combined stimulation therefore creates a synergistic pro-fibrotic environment that models central mechanisms thought to contribute to PVR pathogenesis. We further highlight that nicotinamide attenuates both TGFβ2/Smad- and TNF/NF-κB–associated responses in our system, suggesting a dual inhibitory action across multiple signaling axes.

The revised text is included in the Discussion, stating:

“Mechanistically, TGFβ2 and TNF activate distinct but converging signaling cascades that jointly drive EMT and fibrosis in RPE cells. TGFβ2 primarily signals via Smad2/3 phosphorylation, promoting transcription of extracellular matrix components and EMT regulators, whereas TNF activates NF-κB and MAPK pathways, enhancing inflammatory gene expression and amplifying fibrotic responses. Their combined stimulation therefore provides a synergistic pro-fibrotic environment that reflects key aspects of cytokine-driven mechanisms implicated in PVR. Our data demonstrate that nicotinamide effectively attenuates both TGFβ2/Smad- and TNF/NF-κB–associated responses, suggesting that its inhibitory effect extends across multiple signaling axes. This dual blockade may underlie the robust suppression of EMT markers, contractility, and VEGFA secretion observed in our model.”

We believe that these revisions have significantly improved the clarity and completeness of our manuscript. We thank the reviewer for their valuable input and hope that the revised version meets the journal’s standards for publication.

Sincerely,

Heiko Fuchs

on behalf of all co-authors

---

## [Decision Letter · Decision Letter 1]

24 Sep 2025

Dear Dr. Fuchs,

Thank you for submitting your revised manuscript to PLOS ONE. The reviewers are supportive of your work, but still had some concerns that you should address to the best of your ability. Therefore, we invite you to submit a second revision of your manuscript that adequately addresses the points they raised. Requests for substantial additional experimentation that are not feasible can be addressed by acknowledging limitations in the current research design.

We look forward to receiving your revised manuscript.

Kind regards,

Andre van Wijnen

Academic Editor

PLOS ONE

Journal Requirements:

Reviewers' comments:

Reviewer's Responses to Questions

**Comments to the Author**

Reviewer #1: (No Response)

Reviewer #2: (No Response)

Reviewer #3: (No Response)

2. Is the manuscript technically sound, and do the data support the conclusions?

Reviewer #1: Yes

Reviewer #2: Partly

Reviewer #3: Yes

3. Has the statistical analysis been performed appropriately and rigorously?

Reviewer #1: Yes

Reviewer #2: No

Reviewer #3: Yes

4. Have the authors made all data underlying the findings in their manuscript fully available?

Reviewer #1: Yes

Reviewer #2: Yes

Reviewer #3: Yes

5. Is the manuscript presented in an intelligible fashion and written in standard English?

Reviewer #1: Yes

Reviewer #2: Yes

Reviewer #3: Yes

Reviewer #1: The author mentioned “in methods section to explicitly state: “After transfer, total protein bands were visualized using the ‘Stain-Free blot’ option of the ChemiDoc™ Imaging System following 45 s UV activation and automatic exposure acquisition.” However�the corresponding raw stain-free total protein blots have not been included in the Supplementary Figures SF3–SF6. Please the author check and add or label raw stain-free total in the Supplementary Figures SF3–SF6.

Reviewer #2: comments uploaded as an attachment. Please see full comments from attachment

Comments to the authors

Authors have investigated PVR related EMT using ARPE-19 cell line and TGF-β and TNF-α induced EMT model. They have also tested nicotinamide potential to prevent EMT.

Abstract

-TNT is confusing and the text does not tell what it means. Please revise and make easy for reader to get meaning already from abstract.

Material and methods

- Need to fulfill reagent information e.g. supplier country information

- western blot line 177, complete cells were treated.

-add species of antibodies e.g. FN1.

- marking of the sample amount is confusing, lines 208-209 . Total sample 9? Please clarify into text.

- Add also concentration information to TGF-β and TNF-α.

- Why ANOVA was used even the sample amount was so small?

Results

- Make figures bigger and more visible.

- Supplementary video 1. Why TNT cell layer is damaged at the end? Is concentrations too high? In which timepoint other measurements have done? Are EMT markers formed due to death cells? Does video present migration? Basicly, when treat cells too high concentrations they always detach from cell plate well. Now figures and supplementary files are separated from text at the end, thus hard to follow exactly.

- Supplementary video SV2. Why are the nuclei accumulated almost on one side of the cell plate well? What it mean and what is the conclusion of it.

- Is sample amount three? Refering supplementary Table related to Fig 6. It is quite small sample amount. Is there done only on repetition with three samples? In figure legend, there should be mentione the sample amount (n).

-It would be interesting to see some comparing epithelial markers changes (e.g. occluding, E-cadherin, or ZO-1) to mesenchymal markers increase (e.g. α.SMA, N-cadherin, fibronectin, vimentin) after all treatments compared to control either in protein or gene expression levels.

- Fig 2, Why nuclei tend to be packed as densely. What does it mean? Make Fig 2 more visible. Now it is quite cloudy for interpret.

- Fig 2 B. Why is velocity scale (µm/min) with colours presented at the below of the figure? The figure does not really present any movement and the velocity at the below does not tell anything. Even colours presented in the velocity scale are not marked or presented in the figure. Please mark or explain velocity meaning or delete it under the figure and from legend. White dots represent aggregates. That should be shown also some concretic way. Now it is only showed by white dots which really not be as result. Formed aggregates should be proven to be formed at that site. It can just also be empty area.......

See rest comments from attachment

Reviewer #3: Summary

This manuscript presents a novel in vitro model of proliferative vitreoretinopathy (PVR) using ARPE-19 cells stimulated with TGF-β2 and TNF-α (termed "TNT"). This dual-cytokine approach successfully induces robust epithelial-mesenchymal transition (EMT), extracellular matrix remodeling, cell migration, collagen contraction, and VEGF-A secretion-key hallmarks of PVR pathophysiology. The authors demonstrate that nicotinamide (NAM) effectively attenuates these fibrotic and pro-angiogenic responses, supporting its therapeutic potential. The model is technically sound, amenable to live-cell imaging, and highly reproducible, offering a valuable platform for future compound screening efforts.

Minor points:

a.) Although the use of ARPE-19 cells is well-justified for establishing this screening model, the Discussion should more prominently acknowledge the inherent limitations compared to primary RPE cells or iPSC-derived RPE, particularly regarding translational relevance. For example, absence of melanin could affect cytokine responses and as as immortalized cells, ARPE-19 may already exhibit mesenchymal features that could overestimate EMT and therapeutic effects.

b.) While there are space limitations, some controls are very important to show in the main manuscript. Thus, including one representative stain-free Western blot in the main figures (alongside the supplementary data) would enhance transparency and allow readers to better assess loading controls and normalization approaches.

**Do you want your identity to be public for this peer review?** For information about this choice, including consent withdrawal, please see our Privacy Policy

Reviewer #1: **Yes: ** Xiaodong chen

Reviewer #2: No

Reviewer #3: No

---

## [Author Response · Author response to Decision Letter 2]

24 Nov 2025

Dear reviewers,

First, we thank the reviewers for taking the time to read and review our manuscript. We have improved the manuscript based on their comments and suggestions and would like to address their points in detail below.

Responses to Reviewer #1 comments

Reviewer #1 - additional comment (stain-free total protein blots):

“The author mentioned in methods section to explicitly state: “After transfer, total protein bands were visualized using the ‘Stain-Free blot’ option of the ChemiDoc™ Imaging System following 45 s UV activation and automatic exposure acquisition.” However, the corresponding raw stain-free total protein blots have not been included in the Supplementary Figures SF3-SF6. Please the author check and add or label raw stain-free total in the Supplementary Figures SF3-SF6.”

Response:

We thank the reviewer for carefully checking this point. In the original submission, the labelling and numbering of the supplementary figures were indeed not fully consistent with the description in the text, and the stain-free total protein images were not clearly identifiable.

In the revised version, we have corrected this and now provide, for each western blot experiment, both the raw immunoblot and the corresponding stain-free total protein blot as separate panels, with explicit labelling and highlighting the numbers of biological replicates. Specifically:

- Supplementary Figure SF1A-B: COL1A1 immunoblot and matching stain-free total protein blot for Figure 1C

- Supplementary Figure SF2A-B: FN1 immunoblot and matching stain-free total protein blot for Figure 1C

- Supplementary Figure SF6A-B: COL1A1 immunoblot and matching stain-free total protein blot for Figure 4C

- Supplementary Figure SF7A-B: FN1 immunoblot and matching stain-free total protein blot for Figure 4C

We have also updated the references to the supplementary material in the main text and Methods section so that the numbering now correctly points to these stain-free total protein controls. All corresponding densitometric data are included in Supplementary Table ST1 (Raw data).

We hope that this revised presentation fully addresses the reviewer’s concern regarding the availability and labelling of the stain-free total protein blots.

Responses to Reviewer #2 comments

Reviewer Comment 1 (Abstract):

“TNT is confusing and the text does not tell what it means. Please revise and make easy for reader to get meaning already from abstract.”

Response:

We thank the reviewer for this helpful remark. We agree that the abbreviation “TNT” should be introduced as clearly as possible in the abstract. We have therefore revised the relevant sentence to explicitly define the combined cytokine stimulus, which now reads as follows:

“Here, we establish a cytokine-driven in vitro PVR model by co-stimulating ARPE-19 cells with transforming growth factor beta 2 (TGFB2) and tumor necrosis factor-alpha (TNF); this combined TGFB2/TNF stimulus is hereafter referred to as ‘TNT’.”

We hope that this revised wording makes the meaning of the abbreviation immediately clear to the reader.

Reviewer Comment 2 (Materials and Methods – reagents):

“Need to fulfill reagent information e.g. supplier country information.”

Response:

We thank the reviewer for this comment and fully agree that transparent reagent information is essential for reproducibility. In the Materials and Methods section, we already provide the catalogue numbers (and, where applicable, the supplier names) for all reagents used. As these catalogue numbers are globally unique and independent of the supplier’s local branch or country, they allow readers worldwide to obtain the identical reagents without ambiguity. We therefore did not further specify supplier country information, as we consider the current level of detail sufficient to ensure reproducibility.

Reviewer Comment 3 (Western blot):

“Western blot line 177, complete cells were treated.”

Response:

We thank the reviewer for this helpful remark and agree that the original wording could be misinterpreted. We have therefore clarified the description in the Methods section to make explicit that intact ARPE-19 cells were treated prior to lysis. The revised sentence now states:

“After 24 h, cells were treated with TGFB2, TNF, TNT, or control with or without NAM for five days before lysis.” This wording should avoid any ambiguity regarding the treatment of complete cells before protein extraction.

Reviewer Comment 4 (Antibodies):

“Add species of antibodies e.g. FN1.”

We thank the reviewer for this helpful suggestion. We have revised the Methods section to explicitly state the host species of the primary antibodies. The respective sentence now reads:

“Membranes were blocked with 5% milk powder (Carl Roth #T145.2) in 1× Tris-buffered saline (TBS) at room temperature (RT) for 1 h, followed by overnight incubation at 4°C with a 1:1000 dilution of Fibronectin 1 (FN1) rabbit mAB (Cell Signaling #26836S) and COL1A1 rabbit mAB (Cell Signaling #72026S).”

Reviewer Comment 5 (Sample amount):

“Marking of the sample amount is confusing, lines 208–209. Total sample 9? Please clarify into text.”

Response:

We thank the reviewer for this comment. In the Materials and Methods section, we specify the replicate structure as follows:“Each condition was analyzed in three technical replicates across three biological replicates.”This indicates that three independent biological experiments were performed (n = 3), with each biological replicate measured in triplicate (technical replicates). To make the total number of biological replicates more immediately transparent for the reader, we now additionally state the number of biological replicates (n) in the respective figure legends and list them in detail in Supplementary Table ST1 (“Raw data”).

Reviewer Comment 6 (Concentrations) “Add also concentration information to TGF-β and TNF-α.”

Response:

We thank the reviewer for this helpful suggestion. We have already clarified the cytokine concentrations in the Materials and Methods section. The text states that ARPE-19 cells were treated with 10 ng/mL TGFB2 and 5 ng/mL TNF, unless otherwise indicated (e.g.SF 3+4).

Reviewer Comment 7 (Statistics):

“Why ANOVA was used even the sample amount was so small?”

Response:

We thank the reviewer for this comment. In our experiments we routinely compared 4–8 different conditions (control, TGFB2, TNF, TNT, with or without NAM). In such settings, a series of pairwise t-tests would markedly increase the risk of type I error, whereas one-way ANOVA is specifically designed to test for differences across multiple groups within a single model. Although the number of biological replicates per group is relatively small (n ≥ 3), this approach is widely used and appropriate for exploratory in vitro studies with multiple treatment conditions.

Reviewer Comment 8 (Results – Figures):

“Make figures bigger and more visible.”

Response:

We thank the reviewer for this helpful suggestion. In the revised manuscript, we have increased the sizes of Figure 2 and Figure 5 to improve visibility and interpretation. We would also like to note that the figures displayed in the review PDF generated by the PLOS ONE submission system are subject to automatic downscaling and compression, which can reduce their apparent resolution. The original figure files that we uploaded are high-resolution images, and in the final typeset version of the article, the publisher will use these full-resolution files so that readers can zoom in without a substantial loss of image quality.

Reviewer Comment 9 (Supplementary Video 1):

“Supplementary video 1. Why TNT cell layer is damaged at the end? Is concentrations too high? In which timepoint other measurements have done? Are EMT markers formed due to death cells? Does video present migration? Basicly, when treat cells too high concentrations they always detach from cell plate well.”

Response:

We thank the reviewer for these questions and for the careful inspection of Supplementary Video 1, and we apologize that an important piece of information was missing in the original legend. In the revised manuscript, we have now clarified that the video shows phase-contrast live-cell imaging of control- and TNT-treated ARPE-19 cells after 72 h of prior exposure to the respective conditions. From this time point onward, TNT-treated cells increasingly migrate and cluster into multicellular, multilayered aggregates with pronounced contractile behavior. The legend of Supplementary Video 1 has been updated accordingly.

The apparent “damage” of the TNT cell layer at later time points is therefore not due to acute cytotoxicity from excessively high cytokine concentrations, but reflects mechanically driven detachment of the monolayer caused by these contractile aggregates. As the aggregates pull on the surrounding sheet, the cell layer folds and partially lifts off the substrate, which is consistent with the membrane-like contraction described in proliferative vitreoretinopathy (PVR). In pilot experiments using propidium iodide (PI) staining, we did not observe an increased frequency of PI-positive nuclei under TNT conditions compared with controls, which is why we did not further pursue this readout in the final experimental series (data not shown). During the imaging period, we do not observe massive loss of cells or widespread nuclear fragmentation that would be expected with extensive cell death.

The cytokine concentrations used in our model (10 ng/mL TGFB2 and 5 ng/mL TNF, unless otherwise indicated) are within the range commonly applied in in vitro PVR/EMT models and were chosen because they robustly induce fibrotic remodeling without causing global cell loss. EMT and fibrotic markers (FN1, COL1A1, ZO-1 and F-actin reorganization) and VEGFA secretion were quantified at defined end points after TNT exposure (e.g. day 5), as specified in the Methods, i.e. at time points when TNT-treated cultures still consist of viable, contracting cell layers and aggregates. Thus, the increased EMT marker levels reflect active fibrotic remodeling rather than artefacts arising from dying cells.

Finally, Supplementary Video 1 indeed documents TNT-induced migration: cells undergo collective migration towards emerging foci, followed by three-dimensional piling-up into contractile aggregates. This behavior is further quantified and visualized in the main figures and supplementary figures, and the video is intended as a qualitative illustration of these TNT-induced migration and aggregation dynamics.

Reviewer Comment 10 (Supplementary Video 2):

“Why are the nuclei accumulated almost on one side of the cell plate well? What does it mean and what is the conclusion of it.”

Response:

We thank the reviewer for this observation. Supplementary Video 2 was recorded with a 4× objective in a fixed field of view (FOV) within a 24-well plate. At this magnification, the camera captures only a small fraction of the total growth surface, not the entire well. The apparent accumulation of nuclei “on one side of the well” therefore reflects local aggregate formation within this limited FOV rather than a global asymmetry of the whole well.

In replicate recordings, multicellular aggregates formed at multiple locations across the well. The lateral accumulation seen in the presented clip is thus a stochastic local event and illustrates the collective migration and clustering behavior induced under TNT conditions, which is consistent with the PVR-like phenotype we aim to model.

Reviewer Comment 11 (Sample size):

“Is sample amount three? Refering supplementary Table related to Fig 6. It is quite small sample amount. Is there done only on repetition with three samples? In figure legend, there should be mentione the sample amount (n).”

Response:

We thank the reviewer for this comment. For the experiments related to Figure 6 (and all other quantitative assays in this study), we performed three to five independent biological experiments. Thus, the data do not originate from a single experiment with three wells, but from three separate experimental runs. We now explicitly state the number of biological replicates (n) in the corresponding figure legends and in Supplementary Table ST1 (“Raw data”), so that the sample size is immediately transparent to the reader.

Reviewer Comment 12 (Epithelial vs. mesenchymal markers):

“It would be interesting to see some comparing epithelial markers changes (e.g. occluding, E-cadherin, or ZO-1) to mesenchymal markers increase (e.g. α.SMA, N-cadherin, fibronectin, vimentin) after all treatments compared to control either in protein or gene expression levels.”

Response:

We appreciate this thoughtful suggestion. Our a priori aim in the present study was to characterize fibrotic remodelling rather than to comprehensively stage EMT, which is why we focused on mesenchymal/fibrotic readouts such as FN1, COL1A1, and F-actin stress fibres that directly report matrix production and contractile reorganization relevant to PVR. We did not perform a broader panel of mesenchymal markers (e.g. α-SMA, N-cadherin, vimentin) or gene expression profiling in this work.

To address the reviewer’s comment, we additionally performed immunofluorescence staining for the epithelial tight-junction marker ZO-1 after five days of treatment. Under our culture conditions, robust junctional ZO-1 staining was only observed in confluent monolayers, which were consistently reached at day 5; therefore, this time point was chosen for epithelial marker analysis. The new data are presented in Supplementary Figure SF8. In control cells, ZO-1 shows a continuous junctional belt, whereas TNT treatment markedly disrupts this membrane-associated pattern. Co-treatment with nicotinamide partially preserves junctional ZO-1 staining compared with TNT alone, consistent with the overall EMT-attenuating and anti-fibrotic effect of nicotinamide in our model.

We did not analyse occludin or E-cadherin in this study, but agree that combining a broader epithelial and mesenchymal marker panel at the protein and/or transcript level would be an interesting objective for future work.

Reviewer Comment 13 (Fig. 2 – visibility & dense nuclei):

“Why nuclei tend to be packed as densely. What does it mean? Make Fig 2 more visible. Now it is quite cloudy for interpret.”

Response:

We thank the reviewer for this comment. As requested, we have enlarged Figure 2 in the revised manuscript to improve visibility. The densely packed appearance of the nuclei is an inherent feature of the TNT condition and reflects the underlying biology of our model: under TNT, ARPE-19 cells migrate and cluster into compact, multilayered aggregates rather than remaining in a simple monolayer. In a single-plane live-cell imaging readout, this three-dimensional piling-up of cells appears as closely apposed or overlapping nuclei within the field of view. Functionally, these multilayered, contractile aggregates are a key hallmark of our in vitro PVR system, as they are required to generate the pronounced tissue contraction and membrane-like behavior observed in Supplementary Video 1.

Reviewer Comment 14 (Fig. 2B – velocity scale & aggregates):

“Fig 2 B. Why is velocity scale (μm/min) with colours presented at the below of the figure? The figure does not really present any movement and the velocity at the below does not tell anything. Even colours presented in the velocity scale are not marked or presented in the figure. Please mark or explain velocity meaning or delete it under the figure and from legend. White dots represent aggregates. That should be shown also some concretic way. Now it is only showed by white dots which really not be as result. Formed aggregates should be proven to be formed at that site. It can just also be empty area in cell plate well for example due to death and detach cells. Should be proven to be aggregate. Otherwise need to be deleted from the figure and legend.”

Response:

We thank the reviewer for these helpful comments on Figure 2B. Regarding the velocity scale, Figure 2B displays nuclear migration tracks generated with the TrackMate/StarDist pipeline. The colour of each track segment encodes the instantaneous nuclear velocity (μm/min) according to the colour bar shown below the panel. In the original submission, this was not explained cle

---

## [Decision Letter · Decision Letter 2]

23 Dec 2025

A TGFB2/TNF-Induced in vitro model of Proliferative Vitreoretinopathy (PVR) using ARPE-19 cells confirms Nicotinamide as an inhibitor of EMT and VEGFA secretion

PONE-D-25-33120R2

Dear Dr. Fuchs,

We’re pleased to inform you that your manuscript has been judged scientifically suitable for publication and will be formally accepted for publication once it meets all outstanding technical requirements.

Kind regards,

Andre van Wijnen, PhD

Academic Editor

PLOS One

Additional Editor Comments (optional):

Reviewers' comments:

Reviewer's Responses to Questions

**Comments to the Author**

Reviewer #1: All comments have been addressed

Reviewer #2: All comments have been addressed

Reviewer #3: All comments have been addressed

2. Is the manuscript technically sound, and do the data support the conclusions?

Reviewer #1: Yes

Reviewer #2: (No Response)

Reviewer #3: Yes

3. Has the statistical analysis been performed appropriately and rigorously?

Reviewer #1: Yes

Reviewer #2: Yes

Reviewer #3: Yes

4. Have the authors made all data underlying the findings in their manuscript fully available?

Reviewer #1: Yes

Reviewer #2: Yes

Reviewer #3: Yes

5. Is the manuscript presented in an intelligible fashion and written in standard English?

Reviewer #1: (No Response)

Reviewer #2: Yes

Reviewer #3: Yes

Reviewer #1: (No Response)

Reviewer #2: Authors have answered all my questions and improved manuscript based on my comments or justify if not. Manuscript is fine to be publish.

Reviewer #3: The authors have provided detailed responses to most of the reviewers’ comments, including improved figure labeling, expanded methodological clarity, and the addition of new data (e.g., ZO-1 staining) and supplementary imaging to support their interpretations. They have appropriately acknowledged the limitations of the ARPE-19 model, clarified the basis for their migration analyses versus cytotoxicity, and strengthened the Discussion to better position this system as a screening platform requiring validation in more physiologic models. Remaining issues are minor and largely editorial, and do not materially affect the scientific rigor or transparency of the work.

**Do you want your identity to be public for this peer review?** For information about this choice, including consent withdrawal, please see our Privacy Policy

Reviewer #1: **Yes: ** Xiaodong chen

Reviewer #2: No

Reviewer #3: No

---

## [Editor Report · Acceptance letter]

PONE-D-25-33120R2

PLOS One

Dear Dr. Fuchs,

I'm pleased to inform you that your manuscript has been deemed suitable for publication in PLOS One. Congratulations! Your manuscript is now being handed over to our production team.

Kind regards,

on behalf of

Dr. Andre van Wijnen

Academic Editor

PLOS One